# Antiviral Activity of Contemporary Contact Lens Care Solutions against Two Human Seasonal Coronavirus Strains

**DOI:** 10.3390/pathogens11040472

**Published:** 2022-04-15

**Authors:** Christiane Lourenco Nogueira, Scott Joseph Boegel, Manish Shukla, William Ngo, Lyndon Jones, Marc G. Aucoin

**Affiliations:** 1Department of Chemical Engineering, University of Waterloo, Waterloo, ON N2L 3G1, Canada; c3lourenconogueira@uwaterloo.ca (C.L.N.); sjboegel@uwaterloo.ca (S.J.B.); 2Centre for Ocular Research & Education (CORE), School of Optometry & Vision Science, University of Waterloo, Waterloo, ON N2L 3G1, Canada; manish.shukla@uwaterloo.ca (M.S.); william.ngo@uwaterloo.ca (W.N.); lyndon.jones@uwaterloo.ca (L.J.); 3Centre for Eye and Vision Research (CEVR), 17W Hong Kong Science Park, Hong Kong

**Keywords:** contact lens solutions, virucidal activity, human coronavirus, HCoV-229E, HCoV-OC43

## Abstract

Background: Given that reports have suggested SARS-CoV-2 can be transmitted via conjunctiva, the ability of contact lens (CL) care products to reduce the infectiousness of two seasonal human coronavirus (HCoV) (HCoV-229E and HCoV-OC43) surrogates for SARS-CoV-2 was investigated. Methods: Biotrue and Boston Simplus (Bausch&Lomb), OPTI-FREE Puremoist and Clear Care (Alcon), and cleadew and cleadew GP (Ophtecs) were tested. Their ability to inactivate HCoV was evaluated using contact times of 4 and 6 h as well as 1% and 10% of virus inoculum. Results: Non-oxidative systems (Biotrue, Boston Simplus, and OPTI-FREE) did not exhibit a significant log_10_ reduction compared to controls for the two viral strains for either incubation time (all *p* > 0.05) when 10% tests were performed. For the 1% test, while Boston Simplus and OPTI-FREE exhibited a significant log_10_ reduction of both HCoV-229E (after 6 h) and HCoV-OC43 (after either 4 or 6 h incubation), those products showed less than 1 log_10_ reduction of the two infectious viruses. Oxidative systems based on hydrogen peroxide or povidone-iodine showed a significant log_10_ reduction compared with the controls for both HCoV-229E and HCoV-OC43 in all tested conditions (all *p* < 0.01). Clear Care led to virus inactivation to below the limit of quantification for tests performed with 1% of inoculum after 6 h incubation, while cleadew and cleadew GP led to inactivation of the two viruses to below the limit of quantification in all tested conditions. Conclusion: Oxidative CL disinfection systems showed significant virucidal activity against HCoV-229E and HCoV-OC43, while non-oxidative systems showed minimal ability to inactivate the HCoV species examined.

## 1. Introduction

In the advent of the coronavirus disease 2019 (COVID-19) pandemic caused by the SARS-CoV-2 virus, the focus on potential means of transmission has been heightened. Although ocular manifestations in patients with COVID-19 seem to be rare [1,2,3,4], some reports have suggested that SARS-CoV-2 can be transmitted via mucous membranes such as the conjunctiva, which can be easily exposed to infectious droplets and fomites during close contact with infected individuals and contaminated hands [5,6,7,8]. Grajewski and colleagues demonstrated specific ACE2 expression in conjunctival epithelial cells, potentially allowing direct entry of SARS-CoV-2 [9]. Therefore, the ocular surface exposed to SARS-CoV-2 might transport the virus to nasal and nasopharyngeal mucosa through the lacrimal duct system, ultimately causing a respiratory tract infection [4]. An in-vivo study showed that rhesus monkeys infected by SARS-CoV-2 via conjunctival inoculation progressed to lung infections [10]. Collectively, these studies indicate that while ocular complications are not a frequent manifestation of coronavirus infections in humans, ocular exposure may represent a potential route of entry for SARS-CoV-2. Since it is known that the virus can be transferred by hand contact, it has been suggested that there may be an increased risk of contracting COVID-19 through contact lens (CL) wear, mainly during their application and removal [11,12].

It is estimated that 175 million people worldwide use CL for vision correction and the majority of these use CL that are reused for periods ranging from two to four weeks prior to replacement [11,13]. Reusable CL must be cleaned, rinsed and disinfected daily using an approved CL care product, and these products can be broadly classified into two main groups: non-oxidative and oxidative systems [14].

Approximately two-thirds of CL users disinfect their daily lens overnight using non-oxidative systems [15]. The simplicity, convenience, and low cost of these solutions probably explain the high frequency of their use [16]. These products typically contain non-oxidative antimicrobials such as polyhexamethylene biguanide (PHMB), polyaminopropyl biguanide (PAPB), myristamidopropyl dimethylamine (Aldox™), and polyquaternium-1. Lenses stored in these solutions do not require a neutralization step before re-insertion onto the cornea. Among the oxidative systems, products comprising hydrogen peroxide are mostly used, which have a long history of safety and efficacy [17]. Another oxidative system based on povidone-iodine is available in Japan, Europe, New Zealand, and Hong Kong [18]. Lenses exposed to these products require neutralization of the oxidizing agent before re-insertion onto the eye. Hydrogen peroxide is commonly neutralized using a platinum disc present in specially designed cases, while povidone-iodine is neutralized by adding a tablet containing a neutralizing agent such as ascorbic acid [19].

All CL care products need to demonstrate that they meet a minimal established antimicrobial activity against bacteria and fungi standard strains, as described by “ISO 14729:2001. Ophthalmic optics. Contact lens care products. Microbiological requirements and test methods for products and regimens for hygienic management of contact lenses” [20]. To date, this requires that a disinfecting CL solution meet the primary stand-alone criteria in which an initial concentration of 10^6^ CFU/mL of bacteria (each of *Serratia marcescens*, *Pseudomonas aeruginosa*, *Staphylococcus aureus*) and fungi (each of *Candida albicans*, *Fusarium solani*) are reduced by mean values of not less than 3 log_10_ (99.9%) and 1 log_10_ (90%), respectively, within the manufacturer’s recommended disinfection time and conditions. Minimal requirements regarding the virucidal activity of CL care products have not been established. Moreover, limited data are available surrounding the efficacy of these solutions against viruses. In view of this lack of data and the potential conjunctival transmission of SARS-CoV-2, the purpose of this study was to assess the efficacy of various soft and rigid CL care products against two human seasonal coronavirus surrogates for SARS-CoV-2, namely HCoV-229E and HCoV-OC43. Both HCoV-229E and HCoV-OC43 are considered Risk Group 2 pathogens, while SARS-CoV-2 is classified as a Risk Group 3 human pathogen. Research with Risk Group 2 pathogens avoids the added costs and biosafety concerns that accompany work with Risk Group 3 pathogens, which require a biosafety level 3 lab [21]. Therefore, a surrogate with the lowest risk level is preferred when undertaking in vitro testing. HCoV-229E is an alphacoronavirus, while HCoV-OC43 is a betacoronavirus (like SARS-CoV-2). Both HCoV-229E and HCoV-OC43 cause mild upper respiratory tract infections (the common cold), while SARS-CoV-2 causes severe lower respiratory tract infection. Moreover, HCoV-OC43 has been linked as the cause of the last big coronavirus pandemic (Russian Flu 1889–1891) [22,23]. Although there are clear differences in the pathogenicity of these viruses, they are in the same virus family, have very similar structures, and are both human respiratory pathogens, making them ideal surrogates for testing against SARS-CoV-2 [24,25,26,27,28] while maintaining greater safety for personnel undertaking the experiments.

## 2. Results

For both 1% and 10% tests, Biotrue, Clear Care, and cleadew did not show cytotoxic effects to both MRC-5 and HCT-8 cells, while Boston Simplus, OPTI-FREE Puremoist, and cleadew GP showed cytotoxic effects to those cell lines. It was verified that 1:10 dilution with cold cell culture medium was efficient to avoid cytotoxic effects caused by Boston Simplus and OPTI-FREE Puremoist to the host cell lines. Therefore, after 4 h or 6 h of incubation, samples containing Boston Simplus and OPTI-FREE Puremoist were diluted at 1:10. Samples containing the other CL care products were also diluted at 1:10 to obtain the same experimental conditions. The unique exception was cleadew GP, where 1:100 dilution was needed to avoid cytotoxic effects to host cell lines. Due to these additional dilutions, the limit of quantification of the assay was 7.9 × 10^1^ PFU/mL for all CL care products, except for cleadew GP, which was 7.9 × 10^2^ PFU/mL. The neutralization controls (Table 1) confirmed that all CL care products were efficiently neutralized and non-toxic to the cells and virus after the incubation period and the performance of the dilutions described above, since the difference in virus titers between all neutralization controls and the negative control were lower than ≤0.5 log_10_. 

Figure 1 and Figure 2 show the virucidal activity of CL care products against HCoV-229E and HCoV-OC43, respectively. For tests performed with 10% of inoculum, the non-oxidative disinfecting systems (Biotrue, Boston Simplus, and OPTI-FREE Puremoist) did not exhibit a significant log_10_ reduction compared to the controls for either HCoV-229E or HCoV-OC43 after either 4 h or 6 h incubations (all *p* > 0.05). For tests performed with 1% of inoculum, Biotrue also did not exhibit a significant log_10_ reduction compared to the controls for either HCoV-229E or HCoV-OC43 after either 4 h or 6 h incubations (all *p* > 0.05). However, Boston Simplus and OPTI-FREE Puremoist exhibited a significant log_10_ reduction of HCoV-229E compared with the controls after 6 h incubation, showing 0.57 (±0.08) and 0.25 (±0.03) log_10_ reduction, respectively (all *p* < 0.01), indicating that it was able to inactivate 73.35% (±4.80) and 44.13% (±4.46) of the infectious virus (Table 2). For HCoV-OC43, Boston Simplus and OPTI-FREE Puremoist exhibited a significant log_10_ reduction after either 4 h or 6 h incubations (all *p* < 0.01) for tests performed with 1% of inoculum. Boston Simplus showed 0.38 (±0.10) and 0.53 (±0.04) log_10_ reduction after 4 h and 6 h incubations, respectively, indicating that it was able to inactivate 58.35% (±10.20) and 70.76% (±2.34) infectious virus. OPTI-FREE Puremoist showed 0.43 (±0.19) and 0.56 (±0.16) log_10_ reduction after 4 h and 6 h incubations, respectively, indicating that it was able to inactivate 62.51% (±17.93) and 72.24% (±9.35) of the infectious virus (Table 3).

In contrast, for tests performed with either 10% or 1% of inoculum, the three oxidative disinfecting systems (Clear Care, cleadew, and cleadew GP) each showed a significant log_10_ reduction compared with the controls for both HCoV-229E and HCoV-OC43 after 4 h and 6 h incubation (all *p* < 0.01). 

For tests performed with 10% of HCoV-229E inoculum, Clear Care showed 1.84 (±0.29) and 2.57 (±0.55) log_10_ reduction of the infectious virus after 4 h and 6 h incubations, respectively, indicating that it was able to inactivate 98.30% (±0.82) and 99.52% (±0.38) of HCoV-229E. For tests performed with 1% of inoculum, Clear Care showed 1.90 (±0.9) log_10_ reduction of HCoV-229E after 4 h incubation, indicating that it was able to inactivate 99.18% of the infectious virus. However, after 6 h incubation, Clear Care led to inactivation of HCoV-229E to below the limit of quantification, providing more than 2.08 log_10_ reduction (Table 2).

For tests performed with either 10% or 1% of inoculum, cleadew and cleadew GP led to inactivation of HCoV-229E to below the limit of quantification. For 10% testing, cleadew and cleadew GP provided more than 3.32 and 2.32 log_10_ reduction, respectively, after 4 h and 6 h of incubation, indicating that it provided more than 99.95% and 99.52% reduction of HCoV-229E. For tests performed with 1% of inoculum, cleadew and cleadew GP provided more than 2.09 and 1.09 log_10_ reduction, respectively, after 4h of incubation, indicating that they provided more than 99.18% and 91.79% reduction of HCoV-229E. Similar results were obtained after 6 h incubation, in which cleadew and cleadew GP provided more than 2.08 and 1.08 log_10_ reduction, respectively, indicating that they provided more than 99.17% and 91.70% reduction of HCoV-229E (Table 2). Additionally, for 10% testing, both cleadew and cleadew GP showed a significantly higher disinfection efficacy against HCoV-229E than Clear Care after 4 h and 6 h of incubation (all *p* ≤ 0.001). In contrast, for 1% testing, cleadew and cleadew GP showed a significantly higher disinfection efficacy against HCoV-229E than Clear Care only after 4 h (both *p* = 0.03).

Regarding HCoV-OC43, for tests performed with either 10% or 1% of inoculum, the oxidative disinfecting systems (Clear Care, cleadew, and cleadew GP) led to virus inactivation to below the limit of quantification after both 4 h and 6 h of incubation. For tests performed with 10% of inoculum, after 4 h incubation, Clear Care and cleadew each provided more than 3.02 log_10_ reduction, indicating infectious virus titer reductions of greater than 99.9% each, whereas cleadew GP provided more than 2.02 log_10_ reduction, corresponding to an infectious virus reduction of greater than 99.1%. After 6 h of incubation, Clear Care and cleadew provided more than 3.23 log_10_ reduction, while cleadew GP provided more than 2.23 log_10_ reduction. Therefore, Clear Care and cleadew provided a decrease in infectious virus greater than 99.9%, while cleadew GP provided an infectious virus titer reduction greater than 99.4%. For tests performed with 1% of HCoV-OC43 inoculum, Clear Care and cleadew each provided more than 2.40 log_10_ reduction, indicating infectious virus titer reductions of greater than 99.6% each, whereas cleadew GP provided more than 1.40 log_10_ reduction, corresponding to an infectious virus reduction of greater than 95.9%. After 6 h of incubation, Clear Care and cleadew provided more than 2.35 log_10_ reduction, while cleadew GP provided more than 1.35 log_10_ reduction. Therefore, Clear Care and cleadew provided a decrease in infectious virus greater than 99.6%, while cleadew GP provided an infectious virus titer reduction greater than 95.5% (Table 3). 

## 3. Discussion

Daily wear, reusable CLs require appropriate overnight cleaning, rinsing, and disinfecting to permit safe lens wear upon reinsertion. Data on the efficacy of CL care products against viruses are limited and the disinfecting efficacy of these products against human coronaviruses has never been reported, with one recent report describing the efficacy of contemporary care systems against a murine coronavirus [29].

This study successfully examined the disinfecting efficacy of six commercially available CL care products against HCoV-229E and HCoV-OC43, two seasonal coronavirus strains that are acceptable surrogates for SARS-CoV-2 [25,26,27,28]. Since SARS-CoV-2 belongs in the same family as HCoV-229E and HCoV-OC43, the results of this study would very likely be applicable to SARS-CoV-2. Coronaviruses are enveloped viruses, meaning that the viral capsid is surrounded by a lipoprotein outer layer, which can be disrupted by disinfection agents through lipid solubilization, membrane disruption, or damage to embedded glycoproteins [30]. Therefore, the use of efficient disinfection agents can lead to coronavirus inactivation, lowering the number of infectious viruses and, consequently, the risk of transmission and infection.

The International Standard ISO 14729 provides a methodology for the evaluation of CL disinfecting systems against bacteria and fungi, but it does not cover virucidal testing and does not include a methodology for viral testing. In comparison, ASTM E1052-20 provides a methodology to test the virucidal activity of disinfectants with viruses in suspension, but does not include recommendations for CL care products. Therefore, this study used both tests to evaluate the virucidal activity of CL care products, one adapted from ISO 14729 (using 1% of inoculum) and the other from ASTM E1052-20 (using 10% of inoculum). 

Prior to testing the effect of CL care products on HCoV-229E and HCoV-OC43, cytotoxicity and neutralization studies were performed. A cytotoxicity control is important to evaluate the toxicity of the CL care products to the host cells used in the end-point dilution assay (EPDA). A solution that is cytotoxic leads to changes in cell morphology, which could be mistaken to be cytopathic effects caused by virus infection, leading to false positive results. According to ASTM International Standard E1052-20 [31], reduction in cytotoxicity can be achieved through additional dilution or gel filtration. In this study, Boston Simplus, OPTI-FREE Puremoist and cleadew showed cytotoxic effects to both MRC-5 and HCT-8 cells. It was verified that an additional 1:10 dilution with cold cell culture medium before the performance of the EPDA was efficient to avoid those effects caused by Boston Simplus and OPTI-FREE Puremoist, while 1:100 dilution was needed to avoid cleadew GP cytotoxicity effects to host cell lines.

Following the validation of cytotoxicity controls, neutralization controls were performed. The neutralization control is important to ensure that the virucidal properties of the CL care products were effectively neutralized and the neutralized solution is non-toxic to both the challenge virus and the host cell line. According to ASTM International Standard E1052-20 [31], neutralization can be achieved through dilution or the use of a chemical neutralizer agent. Among the CL care products included in this study, those based on non-oxidative systems (Biotrue, Boston Simplus and OPTI-FREE Puremoist) do not contain a neutralizer agent in their formulations. However, the products based on an oxidative system (Clear Care, cleadew and cleadew GP) do have a neutralizing agent of some form. The case supplied with Clear Care contains a platinum disk that fully neutralizes the hydrogen peroxide. The povidone-iodine present in both cleadew and cleadew GP is neutralized by ascorbic acid and sodium sulfite, respectively, present in the tablet supplied with these products. The neutralization controls were performed as described in Section 4.4.2. Briefly, the resulting cytotoxicity control was diluted 1:10, except those containing cleadew GP, which were diluted 1:100 (for reasons explained above). Then, HCoV-229E or HCoV-OC43 were added to these dilutions. At this point, it is expected that all CL care products are neutralized (by the neutralizer present in their formulations or by dilution), and therefore comparable levels of infectious virus must be recovered from the negative control (D-PBS) and the tested solutions. As shown in Table 1, the difference between negative control (D-PBS) and all neutralization controls were ≤0.5 log_10_ steps, excluding that residual CL care products could inhibit virus propagation and lead to false positive results. Therefore, the results of neutralization control testing demonstrated that the CL care products were completely neutralized, validating the protocols, which were subsequently used to evaluate the ability of disinfection CL care products in reducing the viral load of HCoV-229E and HCoV-OC43. 

In this study, for tests performed with 10% of inoculum, the non-oxidative disinfecting systems (Biotrue, Boston Simplus, and OPTI-FREE Puremoist) did not exhibit a significant log_10_ reduction compared to controls for either HCoV-229E or HCoV-OC43. However, for tests performed with 1% of inoculum, Boston Simplus and OPTI-FREE Puremoist exhibited a significant log_10_ reduction of HCoV-229E compared with control after 6 h incubation, while those CL care products showed a significant log_10_ reduction of HCoV-OC43 after both 4 h and 6 h incubation. Although the CL care products based on non-oxidative systems showed a reduction in the number of infection virus particles, HCoV-229E and HCoV-OC43 were still recovered after the incubation period. These results demonstrated that the virucidal activity of non-oxidative systems was not potent enough to entirely eliminate the number of infectious viruses. However, for both HCoV-229E and HCoV-OC43, the three oxidative disinfection systems (Clear Care, cleadew, and cleadew GP) showed a significant log_10_ reduction compared with controls and a significantly higher disinfection efficacy than non-oxidative systems for both 1% and 10% tests performed. 

These results are in accordance with studies that have evaluated the efficacy of CL care products against different viruses. Heaselgrave et al. showed that a non-oxidative system containing 0.0001% polyhexamethylene biguanide did not exhibit significant efficacy against herpes simplex virus (HSV) type 1, adenovirus type 8, and poliovirus type 2 [32]. Moreover, Kowalski et al. demonstrated that the disinfection efficacy of 3% hydrogen peroxide (the same concentration present in Clear Care) was significantly higher than a non-oxidative system containing 0.0001% polyaminopropyl biguanide [33]. Both Biotrue and Boston Simplus contain polyaminopropyl biguanide at 0.00013% and 0.0005%, respectively. Recently, Yasir et al. demonstrated that CL care products based on non-oxidative disinfecting systems, including Biotrue, did not kill the mouse hepatitis virus (MHV), a murine coronavirus [29]. In comparison, two products based on oxidative disinfection systems, including cleadew and a product containing 3% peroxide (the same concentration as that used in Clear Care), significantly reduced the numbers of coronaviruses to below the limit of quantification [29]. In addition, several reports have also demonstrated that hydrogen peroxide has higher overall biocidal activity than non-oxidative systems against microbial biofilms, fungi and Acanthamoeba trophozoites and cysts [15,34,35,36,37,38,39,40,41]. 

The biocidal efficacy of hydrogen peroxide is based on the generation of free hydroxyl radicals, which react with lipids, proteins, and nucleic acids, causing denaturation of proteins, disruption of biological membranes, and sulfhydryl bonds in proteins and enzymes [42]. Many viruses are susceptible to hydrogen peroxide, including influenza, rubella, rabies, coronaviruses, and others [43]. In this study, Clear Care was able to inactivate 99.52% of infectious HCoV-229E after 6 h incubation, which is the minimal disinfection time recommended by the manufacturer. Meyers et al. investigated the ability of several oral and nasopharyngeal rinses to inactivate high concentrations of HCoV-229E [44]. Products containing 1.5% H_2_O_2_ as their active ingredient showed a reduction of infectious viruses that ranged from below 90% to 99%. In another study, oral antiseptic rinses containing 1.5% and 3% H_2_O_2_ showed between a 90% and a 99% decrease in infectious SARS-CoV2 [45]. Moreover, in this study, Clear Care led to HCoV-OC43 inactivation to below the limit of quantification, indicating that more than 99.9% of the virus was inactivated, demonstrating that this product is more effective against HCoV-OC43 than HCoV-229E. 

The antimicrobial activity of povidone-iodine is related to the concentration of free molecular iodine, which reacts with mitochondrial enzymes of the respiratory chain and/or cell membrane proteins to inactivate microorganisms. Iodine has been effective against a wide range of viruses, including enteroviruses, polio, herpes, vaccinia, rabies, tobacco mosaic viruses, and coronaviruses [46]. In this study, cleadew and cleadew GP (0.05% povidone-iodine) led to the inactivation of both HCoV-229E and HCoV-OC43 to below the limit of quantification. Moreover, cleadew and cleadew GP showed a significantly higher disinfection efficacy against HCoV-229E than Clear Care. Yasir et al. also demonstrated that cleadew significantly reduced the numbers of murine coronaviruses to below the detection limit [29]. Although using higher concentrations of povidone-iodine than cleadew and cleadew GP, several studies have demonstrated the excellent in-vitro antimicrobial activity of this disinfectant. Povidone-iodine showed high in-vitro efficacy against SARS-CoV and Middle East Respiratory Syndrome (MERS) coronavirus at concentrations as low as 0.23% [47,48]. Oral antiseptic preparations, throat spray, and nasal antiseptics containing 0.5% povidone-iodine have achieved ≥ 99.9% virucidal activity against SARS-CoV-2 [49,50,51]. In addition, a study comparing the antimicrobial efficacy of a povidone-iodine CL disinfection system with a one-step hydrogen peroxide system demonstrated that povidone-iodine is more effective against yeast and *Acanthamoeba* trophozoites [52].

## 4. Materials and Methods

### 4.1. Contact Lens Care Products

Three commercial CL care products based on non-oxidative disinfecting systems (Biotrue, Boston Simplus and Opti-free Puremoist) and three based on oxidative disinfecting systems (Clear Care, cleadew and cleadew GP) were tested in this study. The non-oxidative CL care products chosen represented commonly prescribed solutions that encompassed a variety of preservatives and compositions. Biotrue and OPTI-FREE PureMoist are amongst the most commonly prescribed care products globally and encompass the two most commonly encountered CL biocides (poyquaternium-1 and PHMB). The oxidative systems chosen represented two classes of products that rely on oxidative disinfectants (hydrogen peroxide and povidone-iodine). All products are listed in Table 4, along with their disinfectant agents, and the disinfection time recommended by their manufacturers.

Clear Care is supplied with a special lens case that contains a platinum disc. It is a catalytic disc that gradually reduces the peroxide to water and oxygen over 6 hours of incubation. Tablets are supplied with both cleadew and cleadew GP. The tablet supplied by cleadew contains ascorbic acid (2.0 mg/tablet) and a proteolytic enzyme (0.5 mg/tablet), while the tablet supplied with cleadew GP contains sodium sulfite (2.4 mg/tablet) and a proteolytic enzyme (8.0 mg/tablet). Ascorbic acid and sodium sulfite neutralize povidone-iodine to a safe level within 4 hours of incubation. The solution color change (from orange to colorless) ensures that the appropriate neutralization of povidone-iodine has occurred. The CL case supplied with Clear Care and the tablets supplied with cleadew and cleadew GP were used according to the manufacturer’s instructions to ensure appropriate neutralization occurred in the appropriate case and for the appropriate disinfection time.

### 4.2. Virus and Cell Lines

HCoV-229E (ATCC VR-740) and HCoV-OC43 (ATCC VR-1558) were propagated in MRC-5 (human lung epithelial cell; ATCC CCL-171) and HCT-8 (human ileocecal adenocarcinoma cell; ATCC CCL-244) cells, respectively. MRC-5 cells were maintained in Eagle’s Minimum Essential Medium (EMEM) (Wisent BioProducts, Saint-Jean-Baptiste, QC, Canada) supplemented with 10% Fetal Bovine Serum (FBS), while HCT-8 cells were maintained in RPMI (Wisent BioProducts, Saint-Jean-Baptiste, QC, Canada) supplemented with 10% FBS. Both MRC-5 and HCT-8 cells were maintained in a 5% CO_2_ atmosphere at 37 °C and 100% humidity.

### 4.3. Quantitative Suspension Test and Virus Titration

As described above, to-date no minimum virucidal activity for CL care systems exist and thus no methods to test the virucidal activity of CL care systems have been approved. Thus, two differing quantitative suspension tests were used to determine the virucidal activity of CL solutions in this study. The first one was based on the ASTM International Standard E1052-20 protocol [31], which recommends the use of 10% of inoculum. Briefly, 9 parts of each CL solution were mixed with 1 part of HCoV-229E or HCoV-OC43 (10^6^ MPN/mL). As a control, 9 parts of D-PBS (Dulbecco’s Phosphate Buffered Saline, pH 7.4) were mixed with 1 part of HCoV-229E or HCoV-OC43. The second test was based on ISO 14729 that describes the antibacterial and antifungal testing of CL care products, which recommends that the volume of inoculum does not exceed 1% of the sample volume. Therefore, 99 parts of each CL solution were mixed with 1 part of HCoV-229E or HCoV-OC43 (10^6^ MPN/mL). As a control, 99 parts of D-PBS were mixed with 1 part of HCoV-229E or HCoV-OC43. Triplicates were run for all conditions. All mixtures were incubated for both 4 h and 6 h at room temperature and then were diluted at 1:10 with cold cell culture medium, except for cleadew GP, which was diluted at 1:100 (see results section for reason for this). Ten-fold serial dilutions of samples were made in the corresponding culture medium to assess the virus titer via Tissue Culture Infectious Dose 50 (TCID50) end-point dilution assay [53]. For each dilution step, one row of a 96-well plate containing seeded MRC-5 or HCT-8 cells was inoculated. An extra row of mock-infected cells was included across the bottom as a control. The plates were incubated at 33 °C and 5% CO_2_ for 7 or 14 days for HCoV-229E and HCoV-OC43, respectively. Each well was then scored for the presence or absence of infectious virus by examining for any cytopathic effect (CPE) in the wells (Figure 3) through light microscopy (Zeiss Axiovert 40C, original magnification ×100). The virus titer was assessed using the Most Probable Number [54]. Subsequently, the Log Reduction of virus titer was calculated as the difference between the virus titer after contact with CL solutions and the control.

### 4.4. Controls

Prior to testing the effect of CL care products on HCoV-229E and HCoV-OC43, cytotoxicity and neutralization studies were performed for both 1% and 10% tests.

#### 4.4.1. Cytotoxicity Control

Cytotoxicity tests were performed as single assays using cell culture medium + 2% FBS instead of virus. Although this preparation did not contain virus, the samples were subject to the same procedure that was performed to determine the virus titer (Section 4.3). After incubation of the plates, CPE at each dilution was observed to determine the lowest CL solution concentration at which cytotoxic effects were absent.

#### 4.4.2. Neutralization Control

First, tests were performed exactly as described for cytotoxicity control using cell culture medium + 2% FBS instead of virus. After the incubation period, samples were diluted at 1:10 with cold cell culture medium, except for cleadew GP, which was diluted at 1:100 (see results section for reason for this). Then, 99 parts of those dilutions were mixed with 1 part of HCoV-229E or HCoV-OC43 suspension and incubated for 4 h at room temperature. The virus titre was determined as described above (Section 4.3). For test validation, the difference between negative control (D-PBS) and neutralization controls should be ≤0.5 log_10_ steps, as described in EN 14476:2015 Chemical disinfectants and antiseptics. Virucidal quantitative suspension test for chemical disinfectants and antiseptics used in human medicine [55].

### 4.5. Statistical Analysis

Statistical analysis was conducted using GraphPad Prism v9.2.0. The difference in mean virus titers was tested using a one-way ANOVA with Tukey’s post-hoc multiple comparisons test. The values of virus titers below the limit of quantification were assigned a ‘0’. A *p*-value of 0.05 or below was considered statistically significant.

## 5. Conclusions

The results of this study demonstrated that oxidative CL disinfection systems based on hydrogen peroxide and povidone-iodine have higher virucidal activity than non-oxidative systems against the two viral strains examined, using the test methods employed in this work. Further studies to examine the binding of these viral strains to CL and the potential role of rub-and-rinse on their removal are required to supplement these data regarding the potential difference between these products in the real-world environment [56,57]. In addition, results of CL care products based on other non-oxidative formulations may vary for different formulations not tested in this study and examination of other products is warranted. Finally, since there are no currently accepted standards for testing the efficacy of CL care products against viruses of any sort, a guideline to test virucidal activity of CL care products should be established.

## Figures and Tables

**Figure 1 pathogens-11-00472-f001:**
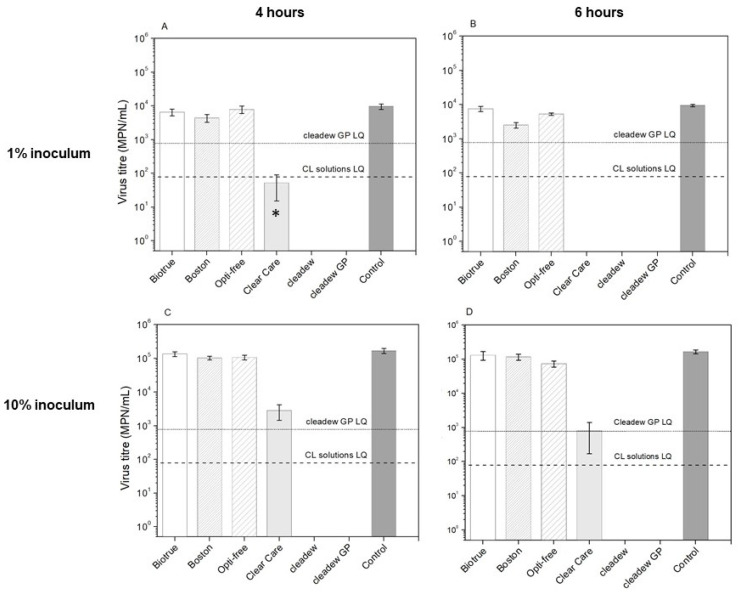
Virucidal activity of CL care products against HCoV-229E after 4 h and 6 h using 1% (**A**,**B**) and 10% inoculum (**C**,**D**) incubation period; plot displays average values and error bars represent standard deviation (n = 3). The minimum disinfection time recommended by the manufacturers is 6 h for OPTI FREE Puremoist and Clear Care, and 4 hours for Biotrue, Boston Simplus, cleadew, and cleadew GP. * In one replicate no virus was detected, while in the other two replicates virus was detected at the limit of quantification. (LQ) limit of quantification.

**Figure 2 pathogens-11-00472-f002:**
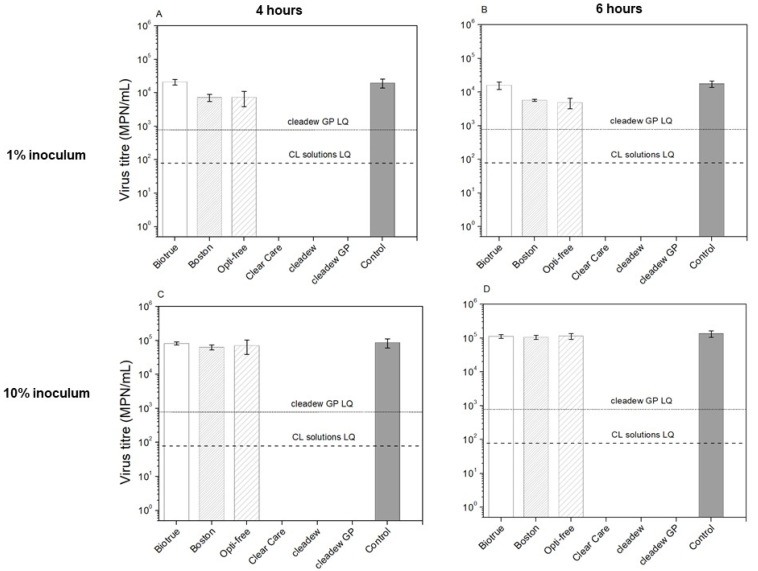
Virucidal activity of CL care products against HCoV-OC43 after 4 h and 6 h using 1% (**A**,**B**) and 10% inoculum (**C**,**D**) incubation period; plot displays average values and error bars represent standard deviation (n = 3). The minimum disinfection time recommended by the manufacturers is 6 h for OPTI FREE Puremoist and Clear Care, and 4 hours for Biotrue, Boston Simplus, cleadew, and cleadew GP. (LQ) limit of quantification.

**Figure 3 pathogens-11-00472-f003:**
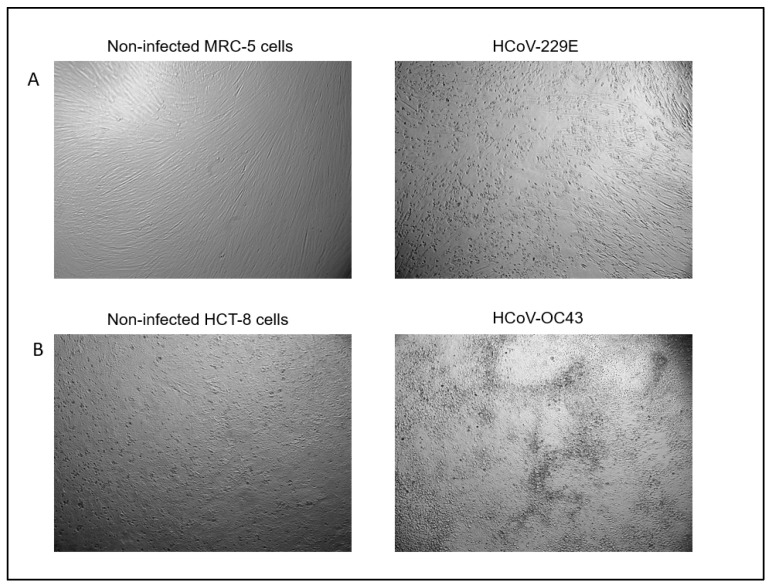
Virus-induced cytopathic effect (CPE) in MRC-5 (**A**) and HCT-8 cells (**B**) after infection with HCoV-229E and HCoV-OC43, respectively. Light microscopy, Zeiss Axiovert 40C, magnification ×100.

**Table 1 pathogens-11-00472-t001:** Reduction in virus infectivity expressed as log_10_ reduction ± standard deviation (n = 3) of neutralization controls.

	1% Test		10% Test	
	HCoV-229E	HCoV-OC43	HCoV-229E	HCoV-OC43
Biotrue	0.04 ± 0.08	0.01 ± 0.09	0.12 ± 0.12	0.09 ± 0.04
Boston Simplus	0.12 ± 0.06	0.05 ± 0.08	0.12 ± 0.02	0.09 ± 0.04
OPTI-FREE Puremoist	0.04 ± 0.06	0.07 ± 0.09	0.10 ± 0.10	0.04 ± 0.00
Clear Care	0.02 ± 0.05	0.05 ± 0.08	0.08 ± 0.04	0.09 ± 0.04
cleadew	0.01 ± 0.04	0.12 ± 0.13	0.08 ± 0.07	0.09 ± 0.04
cleadew GP	0.10 ± 0.04	0.16 ± 0.06	0.03 ± 0.03	0.17 ± 0.04

**Table 2 pathogens-11-00472-t002:** Virucidal activity of CL care products against HCoV-229E after 4 h and 6 h contact time expressed as log_10_ reduction ± standard deviation (n = 3) compared to control.

	1% Test				10% Test			
	4 h	*p*-Value *	6 h	*p*-Value *	4 h	*p*-Value *	6 h	*p*-Value *
Biotrue	0.17 ± 0.10	>0.99	0.10 ± 0.07	0.27	0.09 ± 0.07	0.9868	0.12 ± 0.13	0.9981
Boston Simplus	0.34 ± 0.11	0.95	0.57 ± 0.08	<0.01	0.20 ± 0.06	0.6371	0.16 ± 0.09	0.9894
OPTI-FREE Puremoist	0.08 ± 0.12	>0.99	0.25 ± 0.03	<0.01	0.19 ± 0.07	0.6887	0.36 ± 0.09	0.6705
Clear Care	1.90 ± 0.09	<0.01	>2.08	<0.01	1.84 ± 0.29	<0.0001	2.57 ± 0.55	<0.0001
cleadew	>2.09	<0.01	>2.08	<0.01	>3.32	<0.0001	>3.32	<0.0001
cleadew GP	>1.09	<0.01	>1.08	<0.01	>2.32	<0.0001	>2.32	<0.0001

* *p*-values were calculated based in comparisons between each CL care product and control.

**Table 3 pathogens-11-00472-t003:** Virucidal activity of CL care products against HCoV-OC43 after 4 h and 6 h contact time expressed as log_10_ reduction ± standard deviation (n = 3) compared to control.

	1% Test				10% Test			
	4 h	*p*-Value *	6 h	*p*-Value *	4 h	*p*-Value *	6 h	*p*-Value *
Biotrue	0.03 ± 0.09	>0.99	0.04 ± 0.12	0.99	0.04 ± 0.01	>0.9999	0.08 ± 0.06	0.7541
Boston Simplus	0.38 ± 0.10	<0.01	0.53 ± 0.04	<0.01	0.12 ± 0.07	0.8557	0.10 ± 0.06	0.9989
OPTI-FREE Puremoist	0.43 ± 0.19	0.01	0.56 ± 0.16	<0.01	0.20 ± 0.05	0.9097	0.08 ± 0.08	0.8145
Clear Care	>2.40	<0.01	>2.35	<0.01	>3.02	<0.0001	>3.23	<0.0001
cleadew	>2.40	<0.01	>2.35	<0.01	>3.02	<0.0001	>3.23	<0.0001
cleadew GP	>1.40	<0.01	>1.35	<0.01	>2.02	<0.0001	>2.23	<0.0001

* *p*-values were calculated based in comparisons between each CL care product and control.

**Table 4 pathogens-11-00472-t004:** Contact lens care products included in this study.

Contact Lens Care Product	Manufacturer	Disinfectant Agents	Minimum Disinfection Time (h)
Biotrue	Bausch & Lomb, Rochester, NY, USA	Polyaminopropyl biguanide 0.00013% and polyquaternium 0.0001%	4
Boston Simplus	Bausch & Lomb, Rochester, NY, USA	0.003% chlorhexidine gluconate and 0.0005% polyaminopropyl biguanide	4
OPTI-FREE Puremoist	Alcon, Fort Worth, TX, USA	Polyquad (Polyquaternium-1) 0.001% and Aldox (Myristamidopropyl Dimethylamine)	6
Clear Care	Alcon, Fort Worth, TX, USA	3% hydrogen peroxide	6
cleadew	Ophtecs, Kobe, Japan	0.05% Povidone-iodine	4
cleadew GP	Ophtecs, Kobe, Japan	0.05% Povidone-iodine	4

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
