# Peer review of "Antiviral Activity of Contemporary Contact Lens Care Solutions against Two Human Seasonal Coronavirus Strains"

_pathogens, 2022, doi:10.3390/pathogens11040472_

Round 1
Reviewer 1 Report
The authors describe the ability of contact lens care products to reduce the infectiousness of two seasonal human coronaviruses (HCoV). The topic is very interesting and the manuscript is well organized.
Line 44 the sentence seems to diminish the value of the entire work what is the impact of this paper if “there is no data to support this concern”?
Line 70 why do the authors cite bacterial and fungal if the aim of the work is virus infection?
Line 106 is possible to express the infectivity as LD90 value?
Line 170 “cytotoxicity and neutralization studies were performed” why the authors did not mention it in the result?
Line 266 as the author confirms the protocol used is unused and give some perplexity.
The text should be revised for typos errors
Author Response
Reviewer #1: The authors describe the ability of contact lens care products to reduce the infectiousness of two seasonal human coronaviruses (HCoV). The topic is very interesting and the manuscript is well organized.
- Line 44 the sentence seems to diminish the value of the entire work what is the impact of this paper if “there is no data to support this concern”?
Agreed. The manuscript was adjusted. It now reads:
Collectively, these studies indicate that while ocular complications are not a frequent manifestation of coronavirus infections in humans, ocular exposure may represent a potential route of entry for SARS‐CoV‐2. Since it is known that the virus can be transferred by hand contact, it has been suggested that there may be an increased risk of contracting COVID-19 through contact lens (CL) wear, mainly during their application and removal [11, 12].
- Line 70 why do the authors cite bacterial and fungal if the aim of the work is virus infection?
The manuscript was adjusted. We have added this information to show that minimal requirements regarding antimicrobial activity of CL care products against bacteria and fungi are established, but not against viruses.
The manuscript now reads:
To-date this requires that a disinfecting CL solution meet the primary stand-alone criteria in which an initial concentration of 106 CFU/mL of bacteria (each of Serratia marcescens, Pseudomonas aeruginosa, Staphylococcus aureus) and fungi (each of Candida albicans, Fusarium solani) are reduced by mean values of not less than 3 log10 (99.9%) and 1 log10 (90%), respectively, within the manufacturer’s recommended disinfection time and conditions. Minimal requirements regarding the virucidal activity of CL care products have not been established.
- Line 106 is possible to express the infectivity as 90value?
The authors would like to respond to this comment, but unfortunately it is unclear what the reviewer is requesting. If the question is regarding LD90, it is not applicable to this study since CL care products were only tested in the concentration supplied by the manufactures.
- Line 170 “cytotoxicity and neutralization studies were performed” why the authors did not mention it in the result?
Results of cytotoxicity and neutralization controls are all described in the first paragraph of the Results (section 2).
- Line 266 as the author confirms the protocol used is unused and give some perplexity. The text should be revised for typos errors
This paragraph was reviewed and adjusted.
Reviewer 2 Report
Line 19 - Log10 reduction in viability?
Line 44-45 - What did this report show? It would be useful to provide some more information about why you undertook the study if the report showed no evidence.
Line 69 - 106 CFU / ml? do you mean 1 x 106
Line 80 - The change in biosafety level is not necessarily only associated with the increase consequences of infection. Explanation needs rewording
Line 87 - I would add that OC43 has been linked as the cause of the last big coronavirus pandemic (Russian Flu 1889-1891) - There are a couple of refs that have been published - https://sfamjournals.onlinelibrary.wiley.com/doi/full/10.1111/1751-7915.13889
Line 91 – ISO14729 uses neutralisers and this would overcome the need for increasing the dilutions (Comments for line 326 state neutralisers which are used for ISO contact lens disinfection testing).
Line 99-101 – You have used PFU as the units but have not done a plaque assay and so the units are viral particles per ml.
Line 101 – You have not neutralised the samples and cannot say you have. You have just diluted the samples beyond their toxic levels to the cells lines.
Line 181 – It is not neutralization if you achieve it by dilution – The ASTM standard appears to say to use a neutraliser. Why was this approach not used as it is the standard approach on the contact lens ISO standards.
Line 209 – It is difficult to accept that non oxidative systems do not work when you have diluted them to a 90% concentration.
Line 216 – 218 What level was of viral inactivation was reported in the paper? How does it compare to yours? HSV-1 is an enveloped virus but the other two are not. Did the author get the same result for all 3 viruses?
Line 297 – Is a 1:10 dilution of the virus in its medium into contact lens solution acceptable? In ISO14729 you would add 100ul to 10ml (1:100 dilution) of test disinfectant to ensure that you limit the dilution effect of the disinfectant. Why did you not use a more concentrated viral titre? I cannot access the ASTM standard to confirm if this is acceptable.
Line 299 – You state “Briefly, 9 parts of each CL solution were mixed with 1 part of HCoV-229E 299 or HCoV-OC43” but what was the titre determined to be for each strain?
Line 303-304 – You state the explanation for the Cleadew dilution being 1:100 is stated below but I cannot see an explanation for the difference below (this is because the methods are at the end according to journal structure?)
Line 304-306 – Why did you not use FBS in the medium? The reference at the end of the sentence is just for the calculation of TCID.
Line 309 – Why 33 degrees when the cells were originally grown at 37
Line 326 – What was the neutraliser used prior to cell culture? For PHMB solutions this should be Tween Lecithin. For peroxide this should be catalase and for iodine this should thiosulphate (ISO14729 uses Dey Engley). In the absence of neutralising the peroxide and iodine solutions the inactivation would continue giving the increased inactivation that has been observed. Plus according to ISO 14729 neutraliser controls should have been performed by mixing disinfectant 1:10 with neutraliser to confirm neutralisation.
Author Response
We thank the reviewer for their comments:
- Line 19 - Log10 reduction in viability?
It is Log Reduction of virus titre, that was calculated as the difference between the virus titre after contact with CL solutions and the control (as explained in section 4.3).
- Line 44-45 - What did this report show? It would be useful to provide some more information about why you undertook the study if the report showed no evidence.
The manuscript was adjusted.
- Line 69 - 106 CFU / ml? do you mean 1 x 106
Many thanks for picking this up. This (along with a few other typos) have been corrected.
- Line 80 - The change in biosafety level is not necessarily only associated with the increase consequences of infection. Explanation needs rewording
The manuscript was adjusted.
- Line 87 - I would add that OC43 has been linked as the cause of the last big coronavirus pandemic (Russian Flu 1889-1891) - There are a couple of refs that have been published - https://sfamjournals.onlinelibrary.wiley.com/doi/full/10.1111/1751-7915.13889
Many thanks for this point. Agreed. The manuscript was adjusted and this information was added.
- Line 99-101 – You have used PFU as the units but have not done a plaque assay and so the units are viral particles per ml.
As most probable number (MPN) was used to quantify infectious viral particles, the virus titer unit was modified to MPN/mL.
- Line 209 – It is difficult to accept that non-oxidative systems do not work when you have diluted them to a 90% concentration.
As confirmed in our new experiments undertaken post review of the initial manuscript, non-oxidative systems showed minimal ability to inactivate HCoV species even when tested at 99% concentration. The results of tests using 1% of inoculum were added into the manuscript.
- Line 216 – 218 What level was of viral inactivation was reported in the paper? How does it compare to yours? HSV-1 is an enveloped virus but the other two are not. Did the author get the same result for all 3 viruses?
The results of all studies (including this one) that evaluated anti-viral activity of CL care products were compared with the results of the present study. Although HSV-1 is an enveloped virus, while AV-8 and PV-2 are non-enveloped, approximately 1 log reduction was found against the 3 viruses after 4h exposure with MediCare soft, a CL care product based on non-oxidative disinfection system in a previous study by other authors. Even though different viruses were analyzed, the results are similar to those obtained in this study, indicating that the virucidal activity of non-oxidative systems is not potent enough to eliminate several types of infectious viruses.
- Line 297 – Is a 1:10 dilution of the virus in its medium into contact lens solution acceptable? In ISO14729 you would add 100ul to 10ml (1:100 dilution) of test disinfectant to ensure that you limit the dilution effect of the disinfectant. Why did you not use a more concentrated viral titre? I cannot access the ASTM standard to confirm if this is acceptable.
The International Standard ISO 14729 is intended to provide a methodology for the evaluation of CL disinfecting systems against bacteria and fungi. It does not cover virucidal testing and, therefore, it does not include a methodology for viral testing that we could follow. Therefore, this study was initially performed following the “ASTM International Standard E1052-20 Standard Practice to Assess the Activity of Microbicides against Viruses in Suspension”. As ASTM E1052-20 recommend the use of 10% of inoculum volume, and ISO 14729 recommends 1% (and the reviewers questioned our decision), the authors also performed 1% testing and these results have been added into this revised manuscript.
Regarding the viral titer, all experiments were performed using the highest titer amplified for each coronavirus tested.
- Line 299 – You state “Briefly, 9 parts of each CL solution were mixed with 1 part of HCoV-229E 299 or HCoV-OC43” but what was the titre determined to be for each strain?
The titer of stock suspensions of both HCoV-229E and HCoV-OC43 were 106 PFU/mL. This information was added into the manuscript.
- Line 303-304 – You state the explanation for the Cleadew dilution being 1:100 is stated below but I cannot see an explanation for the difference below (this is because the methods are at the end according to journal structure?)
You can find the explanation in the first paragraph of the Results section: the unique exception was cleadew GP, where 1:100 dilution was needed to avoid cytotoxic effects to host cell lines. Manuscript was adjusted to clarify this point.
- Line 304-306 – Why did you not use FBS in the medium? The reference at the end of the sentence is just for the calculation of TCID.
For End-Point Dilution Assay (EPDA), 10-fold serial dilutions were performed in medium without FBS to prevent bubble formation during the pipetting steps.
- Line 309 – Why 33 degrees when the cells were originally grown at 37
33oC is the temperature recommended by ASTM for HCT-8. This temperature is also used when performing EPDA to slow down the cell growth.
All the answers to questions regarding neutralization are responded to below (please see that section of response to reviewers):
- Line 91 – ISO14729 uses neutralisers and this would overcome the need for increasing the dilutions (Comments for line 326 state neutralisers which are used for ISO contact lens disinfection testing).
- Line 101 – You have not neutralised the samples and cannot say you have. You have just diluted the samples beyond their toxic levels to the cells lines.
- Line 181 – It is not neutralization if you achieve it by dilution – The ASTM standard appears to say to use a neutraliser. Why was this approach not used as it is the standard approach on the contact lens ISO standards.
- Line 326 – What was the neutraliser used prior to cell culture? For PHMB solutions this should be Tween Lecithin. For peroxide this should be catalase and for iodine this should thiosulphate (ISO14729 uses Dey Engley). In the absence of neutralising the peroxide and iodine solutions the inactivation would continue giving the increased inactivation that has been observed. Plus according to ISO 14729 neutraliser controls should have been performed by mixing disinfectant 1:10 with neutraliser to confirm neutralisation.
CL exposed to CL care products based on oxidative systems (ClearCare, cleadew and cleadew GP) require neutralization of the oxidizing agent before re-insertion onto the eye:
- Clear Care is supplied with a special lens case, that contains a platinum disc. It is a catalytic disc that gradually reduces the peroxide to water and oxygen over six hours incubation.
- Tablets are supplied with both cleadew and cleadew GP. The tablet supplied by cleadew contains ascorbic acid (2.0 mg/tablet) and proteolytic enzyme (0.5 mg/tablet), while the tablet supplied by cleadew GP contains sodium sulfite (2.4 mg/tablet) and proteolytic enzyme (8.0 mg/tablet). Ascorbic acid and sodium sulfite neutralize povidone-iodine to safe level within 4 hours of incubation. The solution color change (from orange to colorless) ensures that the appropriate neutralization of povidone-iodine occurred.
As the CL case supplied with ClearCare and the neutralizing tablets supplied with cleadew and cleadew GP were used according to the manufacturer’s instructions, hydrogen peroxide and povidone-iodine were completely neutralized after the incubation period. Therefore, the use of catalase and thiosulphate were not necessary in this study. In fact, ClearCare and cleadew did not show cytotoxic effects on cell monolayers, confirming that hydrogen peroxide and povidone-iodine were completely neutralized after the incubation period. However, cleadew GP showed cytotoxic effects on cells even after the use of the appropriate tablet supplied by the manufacturer. According to the cleadew GP instructions, after the use of the tablet and incubation of 4 hours, the CL must be rinsed with saline solution before re-insertion onto the eye to remove any residual product. There is no recommendation for rinsing CL after disinfection with cleadew. Indeed, cleadew GP was cytotoxic for the cells, while cleadew was not. The cytotoxic effects caused by cleadew would be explained by the presence of proteolytic enzyme (8.0 mg/tablet) in a higher concentration than that present in the cleadew tablet (2.4 mg/tablet).
According to ASTM International Standard E1052-20(1), reduction in cytotoxicity can be achieved through additional dilution or gel filtration. The CL care products that showed cytotoxic effects to both MRC-5 and HCT-8 cells were Boston Simplus, OPTI-FREE Puremoist and cleadew GP, as explained in the first paragraph of the results section. Biotrue, ClearCare (hydrogen peroxide) and cleadew did not show cytotoxic effects to either MRC-5 or HCT-8 cells. For Boston Simplus and OPTI-FREE Puremoist, it was verified that 1:10 dilution with cold cell culture medium was efficient to avoid those effects to the host cell lines. The same dilution was used for all CL care products to have the same experimental conditions for all tested products. The unique exception was cleadew GP, where 1:100 dilution was needed to avoid cytotoxic effects to HCT-8 cells.
As mentioned by the reviewer, ISO 14729 recommends the use of a neutralizer, as Dey-Engley Neutralizing broth (DEB), which would overcome the need of additional dilutions. Serial dilution using DEB was tested by the authors, however it showed toxic effects to both MRC-5 (figure 1) and HCT-8 cells. Therefore, the additional dilutions with cell culture medium were essential in this study to avoid cytotoxic effects to the host cell lines.
See Figure 1: Cytotoxic effect of Dey-Engley Neutralizing broth (DEB) to MRC-5 cells. (A) Serial dilution using EMEM + 10% FBS. (B) Serial dilution using DEB in the attached response.
Although a neutralizer broth was not used, the results reported in Table 2 clearly demonstrated that all CL care products were completely neutralized before the performance of end-point dilution assay. The difference in virus titres between all neutralization controls and the negative control (D-PBS) were lower than ≤0.5 log10. Therefore, the additional dilutions performed with cell medium culture were efficient to avoid an extension of the incubation period, excluding that residual product could inhibit virus propagation and lead to false positive results. The use of cell culture medium as a neutralizer has been reported in other studies that evaluated virucidal activity of chemical disinfectants (2-5).
References
- International A. ASTM E1052-20, Standard Practice to Assess the Activity of Microbicides against Viruses in Suspension. West Conshohocken; 2020.
- Meyers C, Robison R, Milici J, Alam S, Quillen D, Goldenberg D, et al. Lowering the transmission and spread of human coronavirus. J Med Virol. 2021;93(3):1605-12.
- Rabenau HF, Kampf G, Cinatl J, Doerr HW. Efficacy of various disinfectants against SARS coronavirus. J Hosp Infect. 2005;61(2):107-11.
- Statkute E, Rubina A, O’Donnell VB, Thomas DW, Stanton RJ. Brief Report: The Virucidal Efficacy of Oral Rinse Components Against SARS-CoV-2 In Vitro. BioRxiv. 2020.
- Meister TL, Bruggemann Y, Todt D, Conzelmann C, Muller JA, Gross R, et al. Virucidal Efficacy of Different Oral Rinses Against Severe Acute Respiratory Syndrome Coronavirus 2. J Infect Dis. 2020;222(8):1289-92.
- Lambert F, Jacomy H, Marceau G, Talbot PJ. Titration of human coronaviruses, HcoV-229E and HCoV-OC43, by an indirect immunoperoxidase assay. Methods Mol Biol. 2008;454:93-102.
- Talbot PJ, Ekandé S, Cashman NR, Mounir S, Stewart JN. Neurotropism of human coronavirus 229E. Advances in experimental medicine and biology. 1993;342:339-46.
- Schirtzinger EE, Kim Y, Davis AS. Improving human coronavirus OC43 (HCoV-OC43) research comparability in studies using HCoV-OC43 as a surrogate for SARS-CoV-2. Journal of Virological Methods. 2022;299:114317.
- Grossegesse M, Leupold P, Doellinger J, Schaade L, Nitsche A. Inactivation of Coronaviruses during Sample Preparation for Proteomics Experiments. J Proteome Res. 2021;20(9):4598-602.
- Ikner LA, Torrey JR, Gundy PM, Gerba CP. A Continuously Active Antimicrobial Coating effective against Human Coronavirus 229E. medRxiv. 2020:2020.05.10.20097329.
- Ikner LA, Torrey JR, Gundy PM, Gerba CP. Efficacy of an antimicrobial surface coating against human coronavirus 229E and SARS-CoV-2. American Journal of Infection Control. 2021;49(12):1569-71.
- Butot S, Baert L, Zuber S, Elkins Christopher A. Assessment of Antiviral Coatings for High-Touch Surfaces by Using Human Coronaviruses HCoV-229E and SARS-CoV-2. Applied and Environmental Microbiology.87(19):e01098-21.
- Huang L, Gu M, Wang Z, Tang TW, Zhu Z, Yuan Y, et al. Highly Efficient and Rapid Inactivation of Coronavirus on Non-Metal Hydrophobic Laser-Induced Graphene in Mild Conditions. Advanced Functional Materials. 2021;31(24):2101195.
- Rutala WA, Ikner LA, Donskey CJ, Weber DJ, Gerba CP. Continuously Active Disinfectant Inactivates SARS-CoV-2 and Human Coronavirus 229E Two Days After the Disinfectant Was Applied and Following Wear Exposures. Infection Control & Hospital Epidemiology. 2021:1-9.
- Ikner LA, Beck V, Gundy PM, Gerba CP. A Continuously Active Antimicrobial Coating Remains Effective After Multiple Contamination Events. medRxiv. 2020:2020.09.07.20188607.
- Shet M, Hong R, Igo D, Cataldo M, Bhaskar S. In Vitro Evaluation of the Virucidal Activity of Different Povidone–Iodine Formulations Against Murine and Human Coronaviruses. Infectious Diseases and Therapy. 2021;10(4):2777-90.
- Marchesi I, Sala A, Frezza G, Paduano S, Turchi S, Bargellini A, et al. In vitro virucidal efficacy of a dry steam disinfection system against Human Coronavirus, Human Influenza Virus, and Echovirus. Journal of Occupational and Environmental Hygiene. 2021;18(12):541-6.

Reviewer 3 Report
The authors of this study have examined the effects of contemporary contact lens solutions on human coronaviruses. While the study is well conducted, a few points need clarification prior to publication.
- Cytotoxicity control - According to ASTM E1052-20, 9 parts of the test samples need to be added to 1 part of medium to. It seems that the authors have performed 10-fold or 100-fold dilutions of the contact lens solutions in the medium which is the opposite of what is recommended. Please clarify.
- Can the authors comment on their choice of TCID50 assay which is not as sensitive as a plaque assay especially as this would have allowed for a higher LOD for the CleadewGP as well as the use of a validated neutralizer?
Please remove references to non-existent sections 2.3 (line 322) and 2.5 (line 195) from the article.
Author Response
Reviewer #3: “The authors of this study have examined the effects of contemporary contact lens solutions on human coronaviruses. While the study is well conducted, a few points need clarification prior to publication.
- Cytotoxicity control - According to ASTM E1052-20, 9 parts of the test samples need to be added to 1 part of medium to. It seems that the authors have performed 10-fold or 100-fold dilutions of the contact lens solutions in the medium which is the opposite of what is recommended. Please clarify.
The cytotoxicity control was performed exactly as the recommendation of ASTM E1052-20. The procedure is described in the Material and Methods (section 2.4).
According to ASTM International Standard E1052-20(1), reduction in cytotoxicity can be achieved through additional dilution. Boston Simplus, OPTI-FREE Puremoist and cleadew GP showed cytotoxic effects to both MRC-5 and HCT-8 cells, as explained in the first paragraph of the results section. For Boston Simplus and OPTI-FREE Puremoist, it was verified that 1:10 dilution with cold cell culture medium was efficient to avoid those effects to the host cell lines. The same dilution was used for all CL care products to have the same experimental conditions for all tested products. The unique exception was cleadew GP, where 1:100 dilution was needed to avoid cytotoxic effects to HCT-8 cells.
- Can the authors comment on their choice of TCID50 assay which is not as sensitive as a plaque assay especially as this would have allowed for a higher LOD for the CleadewGP as well as the use of a validated neutralizer?
The classical plaque assay appears to be inadequate to titrate HCoV-229E and HCoV-C43 (6, 7). Although it was reported that HCoV-229E infectious titers could be determined by plaque assay on specific cell lines, this proved not to be a reliable assay and, as for HCoV-OC43, alternative assays employing different cell-lines have been used, such as modified plaque assays, indirect immunoperoxidase assay and TCID50 (6, 8).
The American Type Culture Collection (ATCC) recommends the use of MRC-5 (ATCC CCL-171) and HCT-8 (ATCC CCL-244) for the propagation of HCoV-229E (ATCC VR-740) and HCoV-OC43 (ATCC VR-1558), respectively. Therefore, authors decided to use those cell-lines to propagate and titer the tested viruses using EPDA/ TCID50, that is an accepted and commonly used method for titration of HCoV-229E (9-16) and HCoV-OC43 (9, 17).
References
- International A. ASTM E1052-20, Standard Practice to Assess the Activity of Microbicides against Viruses in Suspension. West Conshohocken; 2020.
- Meyers C, Robison R, Milici J, Alam S, Quillen D, Goldenberg D, et al. Lowering the transmission and spread of human coronavirus. J Med Virol. 2021;93(3):1605-12.
- Rabenau HF, Kampf G, Cinatl J, Doerr HW. Efficacy of various disinfectants against SARS coronavirus. J Hosp Infect. 2005;61(2):107-11.
- Statkute E, Rubina A, O’Donnell VB, Thomas DW, Stanton RJ. Brief Report: The Virucidal Efficacy of Oral Rinse Components Against SARS-CoV-2 In Vitro. BioRxiv. 2020.
- Meister TL, Bruggemann Y, Todt D, Conzelmann C, Muller JA, Gross R, et al. Virucidal Efficacy of Different Oral Rinses Against Severe Acute Respiratory Syndrome Coronavirus 2. J Infect Dis. 2020;222(8):1289-92.
- Lambert F, Jacomy H, Marceau G, Talbot PJ. Titration of human coronaviruses, HcoV-229E and HCoV-OC43, by an indirect immunoperoxidase assay. Methods Mol Biol. 2008;454:93-102.
- Talbot PJ, Ekandé S, Cashman NR, Mounir S, Stewart JN. Neurotropism of human coronavirus 229E. Advances in experimental medicine and biology. 1993;342:339-46.
- Schirtzinger EE, Kim Y, Davis AS. Improving human coronavirus OC43 (HCoV-OC43) research comparability in studies using HCoV-OC43 as a surrogate for SARS-CoV-2. Journal of Virological Methods. 2022;299:114317.
- Grossegesse M, Leupold P, Doellinger J, Schaade L, Nitsche A. Inactivation of Coronaviruses during Sample Preparation for Proteomics Experiments. J Proteome Res. 2021;20(9):4598-602.
- Ikner LA, Torrey JR, Gundy PM, Gerba CP. A Continuously Active Antimicrobial Coating effective against Human Coronavirus 229E. medRxiv. 2020:2020.05.10.20097329.
- Ikner LA, Torrey JR, Gundy PM, Gerba CP. Efficacy of an antimicrobial surface coating against human coronavirus 229E and SARS-CoV-2. American Journal of Infection Control. 2021;49(12):1569-71.
- Butot S, Baert L, Zuber S, Elkins Christopher A. Assessment of Antiviral Coatings for High-Touch Surfaces by Using Human Coronaviruses HCoV-229E and SARS-CoV-2. Applied and Environmental Microbiology.87(19):e01098-21.
- Huang L, Gu M, Wang Z, Tang TW, Zhu Z, Yuan Y, et al. Highly Efficient and Rapid Inactivation of Coronavirus on Non-Metal Hydrophobic Laser-Induced Graphene in Mild Conditions. Advanced Functional Materials. 2021;31(24):2101195.
- Rutala WA, Ikner LA, Donskey CJ, Weber DJ, Gerba CP. Continuously Active Disinfectant Inactivates SARS-CoV-2 and Human Coronavirus 229E Two Days After the Disinfectant Was Applied and Following Wear Exposures. Infection Control & Hospital Epidemiology. 2021:1-9.
- Ikner LA, Beck V, Gundy PM, Gerba CP. A Continuously Active Antimicrobial Coating Remains Effective After Multiple Contamination Events. medRxiv. 2020:2020.09.07.20188607.
- Shet M, Hong R, Igo D, Cataldo M, Bhaskar S. In Vitro Evaluation of the Virucidal Activity of Different Povidone–Iodine Formulations Against Murine and Human Coronaviruses. Infectious Diseases and Therapy. 2021;10(4):2777-90.
- Marchesi I, Sala A, Frezza G, Paduano S, Turchi S, Bargellini A, et al. In vitro virucidal efficacy of a dry steam disinfection system against Human Coronavirus, Human Influenza Virus, and Echovirus. Journal of Occupational and Environmental Hygiene. 2021;18(12):541-6.
Round 2
Reviewer 3 Report
The authors have adequately addressed the issues raised by the reviewers.
I did notice a reference to non-existent section 2.3 (Line 399), please fix this.
I have no issues in recommending publishing this article.